

# Self-distillation framework for document-level relation extraction in low-resource environments

Hao Wu, Gang Zhou, Yi Xia, Hongbo Liu and Tianzhi Zhang

Information Engineering University, Zhengzhou, Henan, China

## ABSTRACT

The objective of document-level relation extraction is to retrieve the relations existing between entities within a document. Currently, deep learning methods have demonstrated superior performance in document-level relation extraction tasks. However, to enhance the model's performance, various methods directly introduce additional modules into the backbone model, which often increases the number of parameters in the overall model. Consequently, deploying these deep models in resource-limited environments presents a challenge. In this article, we introduce a self-distillation framework for document-level relational extraction. We partition the document-level relation extraction model into two distinct modules, namely, the entity embedding representation module and the entity pair embedding representation module. Subsequently, we apply separate distillation techniques to each module to reduce the model's size. In order to evaluate the proposed framework's performance, two benchmark datasets for document-level relation extraction, namely GDA and DocRED are used in this study. The results demonstrate that our model effectively enhances performance and significantly reduces the model's size.

## INTRODUCTION

Relation extraction (RE) aims to identify the relations between two entities in a given text, which is essential for information extraction. Nowadays, the main goal of relation extraction is to anticipate the relations between entities in a single sentence by using sentence-level analysis, *e.g.*, *Zhao, Gao & Guo (2023)*, *Zhang et al. (2023)*, *Li et al. (2022)*, *Zheng et al. (2021)*, *Wu et al. (2021)*, *Zhang et al. (2020)*, *Wang et al. (2020)*, *Ye et al. (2020)*, *Yu et al. (2020)*, *Zhang et al. (2018)*; *Zeng et al. (2015)* and *Roberts, Gaizauskas & Hepple (2008)*. Nevertheless, prior research has mostly concentrated on finding relations between single sentences, ignoring the detection of relations between several sentences. In real life, many partnerships are frequently stated in several sentences. According to an analysis of the Wikipedia corpus, at least 40.7% of the relations can only be extracted at the document level (*Yao et al., 2019*). As such, relations must be extracted by models at the document level.

Transformer-based pre-trained models have garnered significant interest in the field of document-level relation extraction recently due to their ability to capture the relation

Corresponding author
Gang Zhou, gzhougzhou@126.com

information between various entities in numerous sentences through an underlying attention mechanism. Nonetheless, the implementation of document-level relation extraction is more challenging than that of sentence-level relation extraction. In particular, it is not possible to deduce the subject and object of a relation from a single sentence alone because they may exist in separate sentences. Second, the model needs to be able to represent entities across sentences because the same entity may be stated in multiple places in the text. Ultimately, to facilitate reasoning, the linkages between certain pairs of entities are frequently discovered through the use of other entities. Determining these multi-hop relations necessitates logical reasoning regarding various entity interactions.

Nonetheless, several current methods improve model performance by simply superimposing representation augmentation modules onto pre-existing models. Although there is some acceleration improvement, the models are not suited for deployment in low-resource situations due to the increased resource consumption caused by these new modules. To solve this problem, we create a self-distillation framework that enhances document-level relation extraction model performance while requiring only a single training session and no additional parameter additions, allowing for deployment in low-resource situations.

This strategy is similar to the multi-exit architectural self-distillation model used in a prior study. In this work, we use ATLOP as a backbone model to provide a framework for knowledge distillation. The entity representation, entity pair representation, and relation classification modules of ATLOP are divided based on their respective functions and distilled individually (*Zhang et al., 2019*; *Lee & Lee, 2021*). More specifically, we include in our comprehensive loss function the contrast loss between the teacher model and the entity and entity pair representations of the backbone model, as well as the contrast loss between the teacher model and the entity pair embedding at the classification layer. Training the teacher model and the backbone model simultaneously enables the former to nearly reach the performance level of the latter in a single training session.

In the Method section of this study, we first introduce individually each module of the overall self-distillation framework. The document-level relation extraction task is defined, and the methods for the encoder module, as well as the entity and entity pair representation enhancement modules, are explained. The Experiments section presents an analysis of the experimental outcomes of the complete self-distillation framework on the DocRED and GDA dataset. Three distinct knowledge distillation tasks and three distinct experimental assessments are used to illustrate the efficacy of the self-distillation framework. Finally, in the Conclusion and Future Works sections, respectively, we provide a summary of our work and outline our plans.

The contributions of our research can be summarized as follows:

- To effectively limit the model size while incorporating new representation enhancement modules, we propose the concept of self-distillation.
- The effectiveness of our self-distillation framework is validated by addressing the multi-hop inference problem, the multi-mention problem, and the evidence sentence problem

individually. Experimental results demonstrate that our self-distillation framework can effectively distill components for various problems.

- Our evaluation of the baseline dataset shows that we can effectively improve the performance of the backbone model in the prediction stage without introducing any additional parameters when compared to the original configuration.

## RELATED WORK

### Document-level relation extraction

In related research on document-level relation extraction, there are currently primarily two methods used. Using the relations between entities, mentions, and sentences to create a graph representation using heuristic rules and dependency structures is a popular method for document-level relation extraction, *e.g.*, *Peng et al. (2017)*, *Liu & Lapata (2018)*, *Christopoulou, Miwa & Ananiadou (2019)*, *Nan et al. (2020)* and *Guo, Zhang & Lu (2019)*. Following that, a graph neural network is used for relation classification. Another common tactic is to utilize the transformer model as an encoder, *e.g.*, *Tang et al. (2020)*, *Wang et al. (2019)* and *Zhou et al. (2021)*. By using self-attention techniques, the transformer model may automatically capture long-distance dependencies, allowing pre-trained models to be used directly in place of explicit graph structure reasoning. As a result, the majority of entity representation techniques used today use the transformer model's self-attention to capture document information. Meanwhile, in order to fully represent entity information and reason effectively about multi-hop relations, more and more scholars have introduced methods in the field of image segmentation into the relation extraction task. For example, the DocuNet model proposed by *Zhang et al. (2021)* redefine the document-level relation extraction task as an entity-pair level classification problem, and predicts entity-level relation matrices through a U-shaped network to capture both local and global information. Similarly, *Li et al. (2021)* who captured global information through co-attention also used a two-dimensional convolutional window to obtain local information in order to capture information of different dimensions between local and global contexts. In addition, *Zhang & Cheng (2022)* modelled inference as a masked network reconstruction problem, where the entity matrix is viewed as an image, which is then stochastically masked and recovered by an inference module in order to capture correlations between relations. The knowledge distillation model proposed by *Tan et al. (2022)* uses two dimensions of axial attention to capture multi-hop relations between entities and achieves good results.

### Self-distillation

Previous studies have primarily concentrated on representing entities and entity pairs, overlooking considerations regarding computing resource utilization and model redundancy. Knowledge distillation (KD) (*Pham et al., 2022*; *Gou et al., 2021*; *Hinton, Vinyals & Dean, 2015*) is a technique that involves transferring learning information from a pre-trained large network (teacher) to a smaller one (student). It is commonly used for model compression, reducing the size of the model while preserving performance. However, the key distinction of self-distillation, a type of knowledge distillation method, is that it simultaneously trains both the teacher model and the student model components

during the training process, but only retains the student model for prediction. Typically, teachers extract knowledge at various levels, such as logits and features (*Romero et al., 2014*), which can be utilized directly or converted into other networks or kernel functions (*Heo et al., 2019a*; *Heo et al., 2019b*; *Kim, Park & Kwak, 2018*). Furthermore, the teacher can consist of a single pre-training network or a group of multiple pre-training networks,or the results of the previous iteration cycle can be used as the teacher model for the next training cycle (*Zhang & Sabuncu, 2020*; *Shen et al., 2022*). The outputs of multiple models, sharing the same structure but employing different initializations, are aggregated (*Simonyan & Zisserman, 2015*). Nevertheless, training multiple models incurs significant computational costs. To enhance the performance of the knowledge distillation model, we eliminate a clear distinction between the roles of teachers and students. Both the teacher model and the student model are trained simultaneously, with shared parameters, thereby reducing resource consumption during the training process.

We apply the concept of self-distillation to the document-level relation extraction work that is being done at the moment. Unlike other self distillation techniques, our approach splits ATLOP backbone model into two modules: the entity representation enhancement module and the entity pair representation enhancement module, based on the features of the document-level relational extraction task. These modules are then independently subjected to self distillation at the same time.

## METHOD

A self-distillation framework for relational extraction at the document level is shown in this section. We classify the conventional document-level relational extraction model into two primary modules: the entity representation module and the entity pair representation module, building upon the current ATLOP. To improve performance, we also add additional modules to the teacher model and train it alongside the backbone model. The diagram for the model is shown in Fig. 1. we describe the task definition of document-level relation extraction. Following that, we present the design of various modules and the loss function of the entire self-distillation framework. We will discuss the two components of the teacher model and the backbone model for the entity and entity pair representation modules respectively.

### Definition

Each document contains a set of entities $\{e_i\}_{i=1}^n$. The task of relation extraction is to identify the relation between entity pairs $(e_s, e_o)$ from the set $\mathcal{R} \cup \{N_A\}$ where $\mathcal{R}$ represents the pre-defined set of relations, and $e_s$ and $e_o$ respectively refer to the subject entity and object entity. Within a document, any entity $e_i$ may appear multiple times in the form of mentions $\left\{m_j^i\right\}_{j=1}^{N_{e_i}}$. If the relation between the entity pair $(e_s, e_o)$ can be represented by any mentioned pair between the two entities, it implies the existence of a relation between them. The label $N_A$ is used to mark the absence of any specific relation between the two entities. During testing, the framework predicts the relation labels for all entity pairs $(e_s, e_o)_{s,o=1...n;s\neq o}$ in the document.
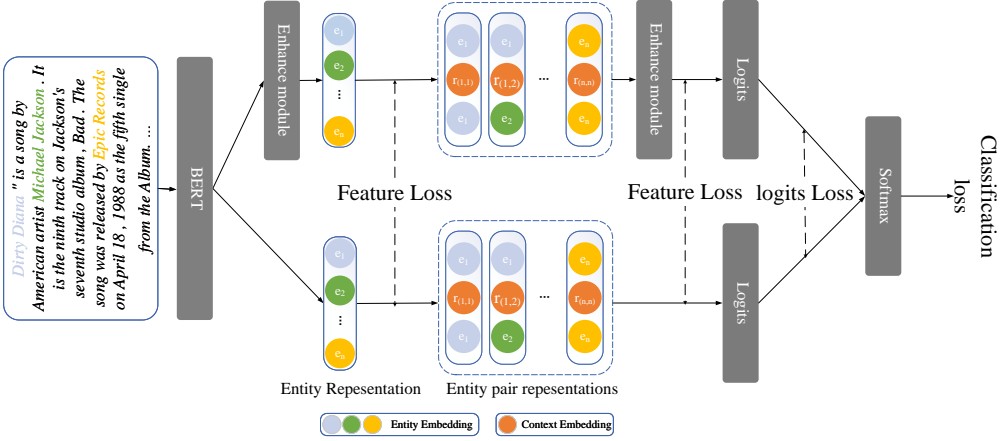

**Figure 1** The overall workflow of the self-distillation framework includes the entity and entity pair representation enhancement modules, along with shared parameters within our framework.

## Encoder module

Considering a given document $d = [x_t]_{t=1}^{l}$, we locate the entity mentions by placing a special symbol "∗" at the start and end of each mention (*Shi & Lin, 2019*; *Soares et al., 2019*; *Zhang et al., 2017*). Subsequently, we encode the document using a pre-trained language model,

$$H = [h_1, h_2, \ldots, h_l] = BERT([\chi_1, \chi_2, \ldots, \chi_l]).$$

Building upon prior research, we employ BERT encoder to encode the document only once, utilizing the same contextual embedding for classification. Here, $h_1$ represents the input corresponding to the embedded word $\chi_1$ in the document. Considering the length limitation imposed by BERT encoder, for documents exceeding 512 tokens, we adopt a dynamic window approach to encode the entire document. Subsequently, we calculate the average of the overlapping embeddings from different windows to obtain the final word embedding.

## Entity representation module

Within the entity representation module, we introduce the entity embeddings both of the backbone model and the teacher model.

### *Backbone model*

The backbone model utilizes the "∗" embedding at the beginning and end of the mention to represent the entity associated with the mention embedding. To obtain the entity's embedding for all mentions $\left\{m_j^i\right\}_{j=1}^{N_{e_i}}$ corresponding to $e_i$, logsumexp pooling (*Jia, Wong & Poon, 2019*) is employed as follows:

$$h_{e_i} = \log \sum_{j=1}^{N_{e_i}} \exp\left(h_{m_j^i}\right).$$

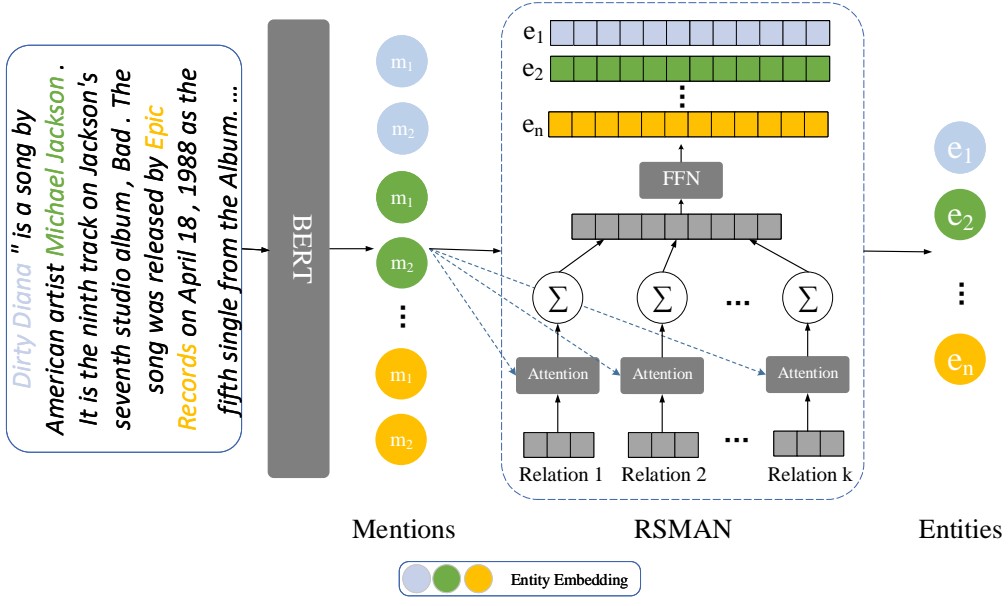

**Figure 2 The workflow of the entity representation enhancement module involves augmenting entity representations using RSMAN.** The entity information is represented based on different relations. Subsequently, we concatenate the entity representations based on their relation types. Lastly, we reduce the dimensionality to match the model's specified dimension using a linear layer.

The pooling operation collects all the information referenced in the document, thereby generating an embedded representation of the entity. Experimental results demonstrate that the algorithm performs superiorly compared to average pooling.

### Teacher model

Various methods for enhancing entity representations can be incorporated into the teacher model to enhance overall model performance, simultaneously the performance of the backbone model can be improved by comparing the entity embedding with that of the teacher model. In comparison to the previous method RSMAN (*Yu, Yang & Tian, 2022*), we improve the performance of the backbone model in the teacher model by employing an attention mechanism to generate distinct entity representations for multiple candidate relations, as shown in Fig. 2.

For each candidate, the parameters of the random initialization can learn $p_r$ as a representation of the relation. Then the representation of the dot product approach is used to calculate the relation $p_r$ and every mention $m_j^i$ between semantic relevance as follows:

$$s_{ij}^r = g\left(p_r, m_j^i\right),$$

where $g$ represents the function used to compute the semantic relevance between the two embeddings. This article employs the dot product to calculate the correlation; however, a multi-layer perceptron (MLP) can also serve this purpose.

Next, the softmax function is utilized to calculate the attention weights for all mentions corresponding to different relations, as shown below:

$$\alpha_{ij}^r = \frac{\exp\left(s_{ij}^r\right)}{\sum_{k=1}^{Q_i} \exp\left(s_{ik}^r\right)}.$$

Once the attention for all entity mentions is obtained, the entity representation for a specific relation can be derived by taking the weighted sum of the attention weights across different relations. For a given relation $r$ between entities $e_i^r$ as illustrated below:

$$e_i^r = \sum_{j=1}^{Q_i} \alpha_{ij}^r m_j^i,$$

$$\grave{h}_{e_l} = \left[e_i^0, e_i^1, \ldots, e_i^r\right].$$

## Entity pairs representation module

In the entity pairs representation module, the entities are embedded in pairs of combinations, for a given entity pair $(h_{e_s}, h_{e_o})$ representing entity $e_s$ and $e_o$ respectively. The embedding of the entity pair is obtained through the basic model and the teacher model, respectively.

### *Backbone model*

In the basic model, the entity embedding is mapped to the hidden state $z$ using linear and nonlinear activation layers, and calculate the probability of relation $r$ with the bilinear function and the sigmoid activation function,

$$z_s = \tanh\left(W_s h_{e_s}\right),$$

$$z_o = \tanh\left(W_o h_{e_o}\right),$$

$$\mathrm{P}(r|e_s, e_o) = \sigma\left(z_s^\top W_r z_o + b_r\right),$$

where the model includes learnable weight matrices $W_s \in \mathbb{R}^{d \times d}$, $W_o \in \mathbb{R}^{d \times d}$, $W_r \in \mathbb{R}^{d \times d}$. The representation of the same entity remains consistent across different entity pairs. To reduce the parameter count in the bilinear classifier, the embedded dimension is divided into $k$ equally-sized groups, and apply the bilinear function and sigmoid activation function within each group. This calculation is used to determine the probability of relation $r$.

$$z_s = \left[z_s^1; \ldots; z_s^k\right],$$

$$z_o = \left[z_o^1; \ldots; z_o^k\right],$$

$$g_i^{(s,o)} = \sigma\left(\sum_{i=1}^k z_s^{i\top} W_r^i z_o^i + b_r\right),$$

$$g^{(s,o)} = \left[ g_1^{(s,o)}, g_2^{(s,o)}, \ldots, g_d^{(s,o)} \right],$$

where the learnable parameters $W_r^i \in \mathbb{R}^{\frac{d}{k} \times \frac{d}{k}}, i = 1 \ldots k$ are utilized, $g^{(s,o)}$ represents the probability of relation $r$ for the entity pair $(e_s, e_o)$. Embedding the groups in $k$ different ways reduces the parameter count from $d^2$ to $d^2/k$.

### Teacher model

In the teacher model, we incorporate various methods to enhance the representation of entity pairs to improve the performance of the model. After the entity representations corresponding to specific relations are arranged in the order of relation types, we employ a linear layer and activation function to map these organized entities to the hidden state $z$, thereby obtaining the representations of the subject and object entities, respectively. The representations of the subject and object entities are as follows:

$$z_S' = \tanh \left( W_s h_e' \right),$$

$$z_o' = \tanh \left( W_o h_e' \right),$$

$$g'^{(s,o)} = \sigma \left( \mathbf{z}'s^\top W_r z'_o + b_r \right),$$

where the learnable parameters $W_s \in \mathbb{R}^{d \times d}, i = 1 \ldots k$ are utilized. $g^{(s,o)}$ represents the probability of relation $r$ for the entity pair $(e_s, e_o)$. Similarly to the backbone model, we utilize $k$ groups to reduce the parameter count. Next, we employ axial self-attention to enhance the neighborhood information along the axes for each entity pair $(e_s, e_o)$. Although previous studies have used axial self-attention to enhance neighbourhood information for relationship classification, we argue that self-attention ignores relationships with samples other than axial. In self-attention, an attention mechanism computes the semantic relevance between the $q$ vector and the $k$ vector, and subsequently applies weights from this attention to the $v$ vector to generate a new feature. In contrast, external attention (*Guo et al., 2022*) functions differently by calculating the relevance between the $q$ vector and a pre-defined common $k$ vector, and subsequently generating a feature map through multiplication of this attention with another externally learnable $v$ vector. Taking this into account, we changed the self-attention in axial attention to an external attention, as shown in Fig. 3.

Axial attention is computed using external attention along the horizontal and vertical axes. A skip connection is added with each calculation along the axis. Given $n \times n$ entity list, for entity pair to $(e_s, e_o)$ by aggregating axial entity pair element $(e_s, e_i)$ and $(e_i, e_o)$ information. The entity pair path traverses two hops between $(e_s, e_i)$ and $(e_i, e_o)$ relations. Furthermore, external attention is used to incorporate information beyond the two hops. Subsequently, this information is utilized to classify the relations between the entity pair $(e_s, e_o)$, encompassing not only the information of the two entities but also incorporating multiple hops. For entity pair $(e_s, e_o)$,

$$r_w^{(s,o)} = r_h^{(s,o)} + \sum_{p \in 1 \ldots n} \text{softmax}_p \left( q_{(s,o)}^T k_{(s,p)} \right) v_{(s,p)},$$

$$r_h^{(s,o)} = g^{(s,o)} + \sum_{p \in 1 \ldots n} \text{softmax}_P \left( q_{(s,o)}^T k_{(p,o)} \right) v_{(p,o)},$$
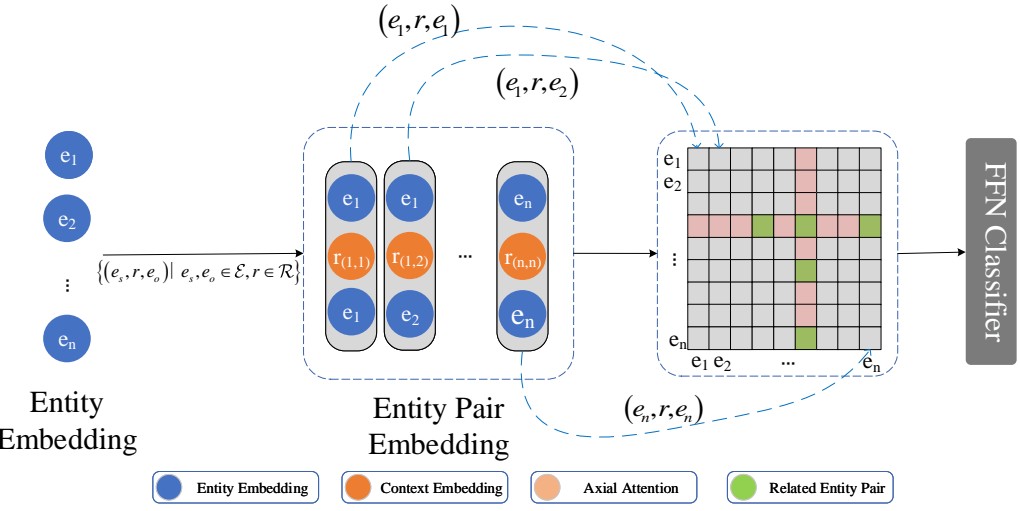

**Figure 3** **Axial attention captures multi-hop information between entity pairs, thereby enhancing their representation.** Upon combining the entity representations in pairs, we utilize axial attention to capture two-hop information represented as an adjacency matrix. The diagram illustrates all the entity pairs that have a two-hop relation with the $(e_4, e_6)$ entity pair. The color red indicates no correlation, while green indicates correlation.

let $q_{(i,j)}, k_{(i,j)}, v_{(i,j)}$ be represented as $W'_q g'^{(i,j)}, W'_k g^{(i,j)}, W'_v g^{(i,j)}$ respectively, where $W'_q \in \mathbb{R}^{d \times d}$, $W'_k \in \mathbb{R}^{d \times d}$, and $W'_v \in \mathbb{R}^{d \times d}$ denote the learnable weight matrices of the model. In contrast to self-attention, $q^T_{(s,o)} k_{(s,p)}$ represents the semantic relevance between the element at position $(s, o)$ and other elements in row $p$. The parameters $k_{(s,p)}$ and $v_{(s,p)}$, which are independent of the input, serve as the weight for the entire training dataset along the horizontal axis.

## Loss

Relation extraction is inherently a multi-class classification task. Traditionally, the relation classification problem has been addressed by employing the cross-entropy loss function. However, this approach is dependent on a universal threshold. To overcome this limitation, we utilize ATL (adaptive threshold loss) as our loss function. We introduce a special category, TH, as an adaptive threshold. Relations exceeding this threshold are labeled as positive examples, while those falling below it are considered negative cases. The formulation of ATL is given by:

$$L_1 = -\sum_{r \in P_T} \log\left(\frac{\exp\left(\text{logit}_r\right)}{\sum_{r' \in P_T \cup \{\text{TH}\}} \exp\left(\text{logit}_{r'}\right)}\right),$$

$$L_2 = -\log\left(\frac{\exp\left(\text{logit}_{\text{TH}}\right)}{\sum_{r' \in N_T \cup \{\text{TH}\}} \exp\left(\text{logit}_{r'}\right)}\right),$$

$$L = L_1 + L_2.$$

The $L_1$ loss considers the positive and TH classes. As there might be multiple positive classes, the overall loss is computed by summing the cross-entropy losses of all positive classes whose logarithmic values exceed that of the TH class. The $L_2$ loss incorporates the negative and threshold classes. This entails a classification of cross-entropy loss, wherein the TH class is explicitly labeled when the logarithm of values is lower than that of the TH class.

Moreover, to facilitate the extraction of valuable information from the teacher model by the backbone model, we incorporate three supplementary losses into our loss function. These losses encompass the embedding contrast loss of the classification layer, the embedding contrast loss of entity features, and the embedding contrast loss of entity pair features between the backbone model and the teacher model. In Parameter sensitivity analysis, we will assess the impact of varying hyperparameters (denoted as $a$ and $b$) on the model's performance.

To calculate the embedding contrast loss for the classification layer, we employ the Mean Squared Error (MSE) loss function, which is defined as follows:

$$L_{logits}^{(s,o)} = \frac{1}{N}\sum_{i=1}^{N}(s_n - o_n)^2.$$

To compute the embedding contrast loss for entity features and entity pair features, we utilize the cosine embedding loss function, which is defined as follows:

$$L_{feature}^{(s,o)} = \begin{cases} 1 - \cos(s,o), & \text{if } y = 1 \\ \max(0, \cos(s,o) - \text{margin}), & \text{if } y = -1 \end{cases}.$$

Thus, our loss function primarily comprises the loss associated with the final relation classification in both the backbone model and the teacher model. Additionally, it incorporates the embedded contrastive loss for the classification layer as well as for the entity and entity pair features, which are defined as follows:

$$L^{(s,o)} = L_{\text{teacher}} + \alpha L_{\text{backbone}} + (1-\alpha)L_{\text{logits}} + \beta L_{\text{feature}},$$

where $\alpha$ and $\beta$ represent hyperparameters. $l_{teacher}$ and $l_{backbone}$ refer to the loss of the teacher model and the backbone model, respectively, obtained through the ATL loss function. The $l_{logits}$ is calculated as the mean square loss between the final logits of the backbone model and the teacher model. $l_{feature}$ represents the cosine similarity computed from the entity features and entity pair features of both the backbone model and the teacher model.

# EXPERIMENTS

## Experiment environment
During our experiments, we use BERT as an encoder to encode the document text. In addition, we use a Linux system with version CentOS Linux release 7.6.1810, containing an Intel (R) Xeon (R) Gold 5320 CPU and a Tesla V100S GPU with CUDA version 11.6. In Table 1, we detail the main Python version, including the Python environment information.

**Table 1  Python environment information.**

| Package | Version |
|---|---|
| python | 3.7 |
| numpy | 1.21.6 |
| pandas | 1.3.5 |
| torch | 1.13.0+cu1163 |
| torchvision | 0.14.0+cu116 |
| transformers | 4.27.4 |
| tokenizers | 0.13.2 |

**Table 2  Statistical analysis of the experimental dataset.**

| Statistics | DocRED |
|---|---|
| distant docs | 101,873 |
| training docs | 3,053 |
| dev docs | 1,000 |
| test docs | 1,000 |
| relations | 97 |
| Avg entities per doc | 19.5 |
| Avg mentions per entity | 1.4 |
| Avg relations per doc | 12.5 |

## Dataset and evaluation metrics

We conduct experiments on DocRED and GDA. The statistical data for DocRED, a large dataset curated especially for relational information extraction from documents, is shown in Table 2. Out of the 3,053 Wikipedia articles in this dataset, about 7% of the entity pairings have more than one relational label.

## Experiment settings

The backbone model we employ is built upon ATLOP, with BERT-base (*Devlin et al., 2019*) serving as the encoder for document encoding in the DocRED (https://github.com/thunlp/DocRED) and GDA (https://bitbucket.org/alexwuhkucs/gda-extraction/src/master/) datasets. For training, we utilize a hybrid precision training model from the Apex library (*Micikevicius et al., 2017*), optimized by AdamW (*Loshchilov & Hutter, 2019*) as the learner, with a learning rate of 5e−5 and linear preheating set to 0.06 (*Goyal et al., 2018*). Additionally, dropout layers with a parameter of 0.1 are inserted between the layers. All hyperparameters are fine-tuned based on the validation set, and the specific values are listed in Table 3.

## Results and analysis

We compare the results of several models on the DocRED and GDA datasets, and the experimental results are presented in Table 4. In the ensuing analyses, we first evaluated the effects of entity representation and entity pair representation enhancement by using three common representational enhancements in ablation experiments. Then, we conducted qualitative results, validity, and parametric sensitivity analyses in the experimental analyses.

**Table 3  Training hyperparameters.**

| Hyperparam | BERT |
|---|---|
| Batch size | 4 |
| Epoch | 30 |
| lr for encoder | 3e−5 |
| lr for classifier | 1e−4 |
| Warmup ratio | 0.06 |
| Num of class | 97 |
| Max seq length | 1,024 |
| Max num of entity | 42 |

**Table 4  Primary results (%) for the development and test sets of DocRED are provided.** The method marked with a star (★) represents the F1 value obtained through five separate training runs with different random seeds. Furthermore, we submitted the best result to CodaLab to obtain the test set results.

| Model | Dev | | Test | |
|---|---|---|---|---|
| | F1 | Ign F1 | F1 | Ign F1 |
| ATLOP | 61.09 | 59.22 | 61.30 | 59.31 |
| SIRE | 61.60 | 59.82 | 62.05 | 60.18 |
| DocuNet | 61.83 ± 0.19 | 59.86 ± 0.13 | 61.86 | 59.93 |
| CorefBERT | 57.51 | 55.32 | 56.96 | 54.54 |
| CorefBERT+RSMAN | 58.24 | 56.26 | 57.53 | 55.30 |
| SSAN | 58.95 | 56.68 | 58.41 | 56.06 |
| SSAN+RSMAN | 59.25 | 57.22 | 59.29 | 57.02 |
| ATLOP- $BERT_{Base}$★ | 60.30 ± 0.11 | 58.32 ± 0.15 | 60.45 | 58.54 |
| Our-RSMAN-Axial★ | 60.86 ± 0.12 | 58.89 ± 0.09 | 60.90 | 58.83 |

## Experimental baseline

We employ RSMAN module to validate the efficacy of distillation in the entity enhancement module and compare the enhancement effects of RSMAN module in two different models. We also include two recent approaches for comparison: the DocuNet model, which captures entity pair dependencies using a U-shaped network inspired by semantic segmentation in the computer vision field, and the SIRE model (*Zeng, Wu & Chang, 2021*), which extracts intra-sentence and inter-sentence relations using sentence-level and document-level encoders, respectively.

Since our axial attention approach considers the relation between an entity and itself, unlike ATLOP backbone model, we need to ensure that the model input contains a square matrix of adjacency matrices comprising all possible entity pairs. To achieve this, we fill in the missing entity pairs in the adjacency matrices during the data construction phase. In order to validate the distillation effect after altering the input data distribution, we retrained ATLOP five times using BERT as the encoder and averaged the results after incorporating the missing entity pairs into the adjacency matrix. In line with the experimental setup, we trained ATLOP backbone model using the parameter settings provided in the article. The

**Table 5 Primary results (%) for the test sets of GDA are provided.** The method marked with a star (★) represents the F1 value obtained through five separate training runs with different random seeds.

| Model | F1 |
|---|---|
| ATLOP-$BERT_{Base}$ | $83.9 \pm 0.2$ |
| Our-RSMAN-Axial w/o distillation★ | 85.7 |
| Our-RSMAN-Axial with distillation★ | 84.72 |

best results on the dev dataset were uploaded to CodaLab (https://codalab.lisn.upsaclay.fr/) to obtain the test dataset results, serving as a baseline for our experiments.

### Evaluation metrics

F1 and Ign F1 are the evaluation measures used in this work. Ign F1 represents the F1 score for relation facts that are excluded from the training, validation, and test sets. It is calculated by taking away the public relations that these sets have in common. Furthermore, in order to obtain the official test results, we forward CodaLab the best results on the development dataset.

### Main results

Recent studies have made extensive use of ATLOP as a generalized backbone as part of their methodology. To improve entity representation and entity pair representation by distillation, we have added RSMAN module and the axial attention module to ATLOP backbone model. The experimental results are denoted by a star (★). Based on the experimental findings, we can see that ATLOP's performance is on par with SSAN (*Xu et al., 2021*), SIRE, and DocuNet. The efficacy of the entity and entity-pair representation enhancement modules based on the self-distillation method is first demonstrated by the observation that ATLOP with self-distillation improves the baseline F1 value by 0.56 (dev set) compared to the baseline F1 value during the model prediction phase.

Furthermore, we examine the lifting effect of SSAN and CorefBERT models simultaneously with the inclusion of RSMAN module. We can see that RSMAN module increase on the CorefBERT and SSAN models is approximately 0.7. Our distillation process, in contrast, successfully lowers the number of parameters without significantly sacrificing performance.

To confirm the effectiveness of the framework, we switched to SciBERT and conducted tests on the GDA dataset in addition to the comparison on the DocRED dataset. The experiment results are shown in Table 5 and show that our strategy is still effective on the GDA biological domain dataset.

In the ensuing analyses, we first evaluated the effects of entity representation and entity pair representation enhancement by using three common representational enhancements in ablation experiments. Then, we conducted qualitative results, validity, and parametric sensitivity analyses in the experimental analyses.

## Ablation study

We do two sets of ablation tests to evaluate the performance of each module in our experiments. We separately disable the entity representation distillation module and

**Table 6  Ablation study on the entity representation enhancement module on the DocRED dataset.**
For the entity enhancement module, we use RSMAN module as a representation of the teacher model enhancement entity. Experimental results show that RSMAN module is effective through our distillation method.

| Model | F1 | Ign F1 |
|---|---|---|
| CorefBERT | 57.51 | 55.32 |
| CorefBERT+RSMAN | 58.24 | 56.26 |
| SSAN | 58.95 | 56.68 |
| SSAN+RSMAN | 59.25 | 57.22 |
| ATLOP | 60.30 | 58.32 |
| Our-RSMAN | 60.70 | 58.89 |

**Table 7  In our backbone model, we exclusively utilized the evidence sentence as the input for the model, while employing the text as the input for the teacher model.** We conducted a comparative analysis of ATLOP's performance when provided with direct input of text and direct input of evidence sentences individually. Experimental results demonstrate that through distillation in our model, a similar effect to the direct input of text can be achieved. This outcome further confirms the efficacy of the enhanced entity representation module in our model.

| Model | F1 | Ign F1 |
|---|---|---|
| ATLOP | 60.30 | 58.32 |
| ATLOP(only with evidence) | 60.17 | 58.19 |
| Our method with evidence | 62.24 | 58.27 |

the entity pair representation distillation module and then evaluate their respective performances against ATLOP backbone model, which serves as the baseline. Our ablation investigations show that both distillation modules provide a substantial contribution to the model's overall performance.

*Entity representation enhanced module*

We test the entity representation improvement module on ATLOP backbone model in order to verify its effectiveness. In particular, we examine the direct effects of RSMAN module on different baseline models and its influence on our model after distillation. RSMAN module shows improvements of 0.73 and 0.3 in CorefBERT and SSAN backbone models, respectively, as shown in Table 6. Additionally, we use the same DocRED dataset to incorporate RSMAN module into ATLOP and conduct distillation, which leads to a 0.39 improvement over ATLOP baseline F1 value. This accomplishment is largely in line with RSMAN module's performance increase.

Moreover, we assess the performance of ATLOP when encoding the entity representation based only on the evidence sentences from the articles, using the original text as input for the teacher model during distillation, in order to offer more proof of the effectiveness of our entity representation enhancement module. Using only the evidence sentences at the prediction step, the findings shown in Table 7 show that our technique achieves a near approximation to the input produced by the original text. This result provides more evidence of the efficacy of our entity representation enhancement.

**Table 8   In the ablation study of the entity pair representation module on the DocRED dataset, we employed axial attention as the method for enhancing the entity pair representation within the teacher model's entity pair representation enhancement module.** The results annotated with a star (⋆) demonstrate an F1 improvement of 0.63 on ATLOP. These outcomes confirm the effectiveness of our model in representing the enhanced entity pairs.

| Model | F1 | Ign F1 |
|---|---|---|
| ATLOP with axial | 0.63⋆ | – |
| Our w/o axial | 60.30 | 58.51 |
| Our with axial | 60.68 | 58.89 |

### Entity pair representation enhanced module

We just concentrate on distilling the representation enhancement module in order to assess the efficacy of our entity pair representation enhancement module. We compare the effect of directly integrating axial attention into ATLOP backbone model with the effect obtained by distillation in our model with respect to the entity pair representation improvement module. When axial attention is given directly to ATLOP, as shown in Table 8, we find that the F1 value increases by 0.63. Furthermore, our model's F1 impact following distillation shows a 0.38 improvement, which is close to the result of adding axial attention directly. This result demonstrates that our entity pair representation improvement module is successful.

## Experimental analyse

This section focuses on the analysis of our model's qualitative results, efficiency, and parameter sensitivity.

### Qualitative results

By reducing the number of parameters, our self-distillation model minimizes the amount of resources used during deployment. In addition to the parameters needed for ATLOP, the additional modules included to improve the entity and entity pair representations can be ignored *via* self-distillation in the model's prediction phase. Moreover, different strategies might be used in place of these two modules.

Since the goal of this study is to utilize knowledge distillation to minimize the number of parameters in the model, we use RSMAN and axial attention modules as examples to show how well our knowledge distillation framework works. We determine the fraction of parameters that are occupied by the axial attention module and RSMAN module, respectively, and the statistics are shown in the Table 9.

Except BERT encoder, which only requires fine-tuning, we computed the parameter requirements for the backbone model, the entity representation enhancement module, and the entity pair representation enhancement module, as shown in Table 9. Specifically, the backbone model ATLOP necessitates approximately 40 million parameters. To enhance the entity representation and entity pair representation, we employ the RSMAN module and the axial attention module, respectively. RSMAN utilizes around 19 million parameters, while the self-attention-based axial attention module requires approximately 17.72 million parameters. On the other hand, the external attention-based axial attention module involves

| Type | Total | Input | Output | Parameters | Sum of participants |
|---|---|---|---|---|---|
| **Backbone Model** | | | | | |
| Linear | 1 | $2 \times 768$ | 768 | $(2 \times 768 \times 768) + 768$ | 1,180,416 |
| Linear | 1 | $2 \times 768$ | 768 | $(2 \times 768 \times 768) + 768$ | 1,180,416 |
| Linear | 1 | $768 \times 64$ | 768 | $(768 \times 64 \times 768) + 768$ | 37,749,504 |
| Linear | 1 | 768 | 97 | $(768 \times 97) + 97$ | 74,593 |
| **RSMAN** | | | | | |
| Linear | 1 | $97 \times 128$ | 768 | $(97 \times 128 \times 768) + 768$ | 9,536,256 |
| Linear | 1 | $97 \times 128$ | 768 | $(97 \times 128 \times 768) + 768$ | 9,536,256 |
| Linear | 1 | 768 | 128 | $(768 \times 128) + 128$ | 98,432 |
| Linear | 1 | 128 | 256 | $(128 \times 256) + 256$ | 33,024 |
| Parameter | 1 | 97 | 256 | $(97 \times 256)$ | 24,832 |
| **Axial attention** | | | | | |
| Linear | 6 | 768 | 768 | $(768 \times 768) + 768$ | 3,543,552 |
| LayerNorm | 6 | 768 | 768 | $(768 + 768)$ | 9,216 |
| **with self-attention** | | | | | |
| Linear | 6 | 768 | 768 | $(768 \times 768) + 768$ | 3,543,552 |
| Linear | 6 | 768 | $2 \times 768$ | $(768 \times 2 \times 768) + 2 \times 768$ | 7,087,104 |
| Linear | 6 | 768 | 768 | $(768 \times 768) + 768$ | 3,543,552 |
| **with extra-attention** | | | | | |
| Linear | 6 | 768 | $768 \times 4$ | $(768 \times 768 \times 4) + 768 \times 4$ | 14,174,208 |
| Linear | 6 | 96 | 32 | $(96 \times 32) + 32$ | 18,624 |
| Linear | 6 | 32 | 96 | $(32 \times 96) + 96$ | 19,008 |
| Linear | 6 | 3072 | 768 | $(3072 \times 768) + 768$ | 14,160,384 |

**Table 9  Number of learnable parameters in the model.**

approximately 31.92 million parameters. Comparatively, the addition of parameters for entity and entity pair representation augmentation accounts for about 47% and 56% of the overall model, respectively.

We can see that the additional representation enhancement modules make up approximately half of the total model proportion. We can reduce the number of parameters by eliminating all of the additional representation enhancement modules after using the knowledge distillation framework to distill the model and save only the backbone model.

### Efficiency analysis

Apart from calculating the required number of parameters, we conducted a comparative analysis of the model's resource utilization during the deployment phase. As depicted in Table 10, saving only the essential parameters resulted in a 47.7% reduction in file size. Moreover, when freshly loading the model onto the graphics card, considering the GPU consumption by the model structure, the model size decreased by 30.9%. Experimental findings demonstrate that the incorporation of additional modules significantly contributes to memory and graphics memory consumption. However, our self-distillation framework helps mitigate these issues to some extent.

**Table 10  Consumption of resources in the forecasting phase.**

| Model | model size (MB) | GPU memory allocated (MB) |
| --- | --- | --- |
| Our with distil | 440 | 1,820 |
| Our w/o distill | 842 | 2,635 |

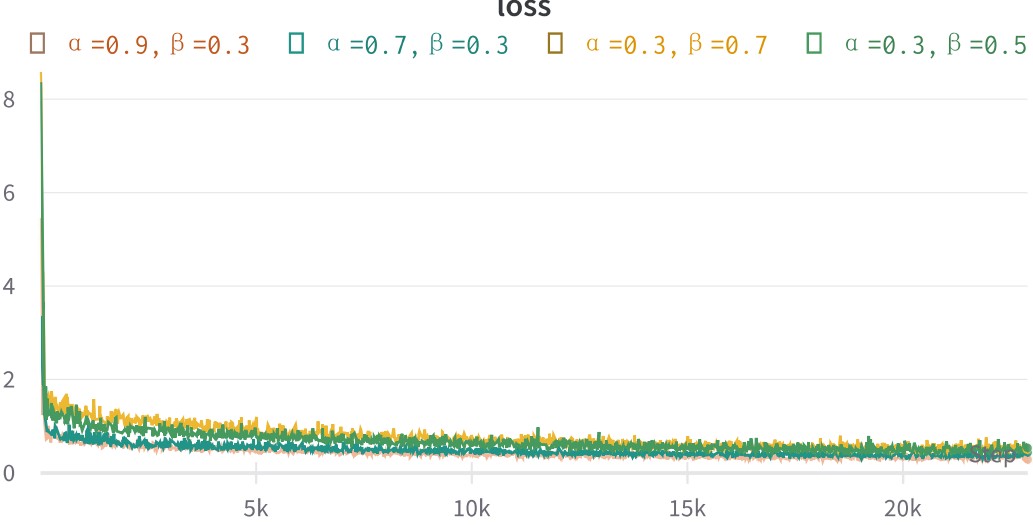

**Figure 4  The graph illustrates how the final loss function of the model varies with the number of iterations for different hyperparameter settings.**

### Parameter sensitivity analysis

In this experiment, we introduce two hyperparameters, $\alpha$, and $\beta$, to ensure that the loss calculation methods remain within an appropriate range when computing the total loss. As a result, we will conduct parameter sensitivity analyses for both $\alpha$ and $\beta$.

By examining the variation of the loss function depicted in Fig. 4, it is evident that the values of $\alpha$ and $\beta$ primarily impact the final loss function of the model during the initial stages. However, as the number of iterations increases, the losses for all modules of the model gradually converge towards a smaller value, rendering the influence of the hyperparameters $\alpha$ and $\beta$ on the model's losses negligible.

The results of our analysis are shown in Fig. 5. The left plot presents the impact of varying the $\alpha$ parameter while keeping the $\beta$ parameter fixed at 0.3 on the model's performance. Conversely, the other plot illustrates the effect of changing the $\beta$ parameter while setting the $\alpha$ parameter to 0.3. The findings indicate that despite arbitrary adjustments to these parameters, the resulting loss values gradually converge to a smaller value, thereby indicating minimal impact on the final model performance.

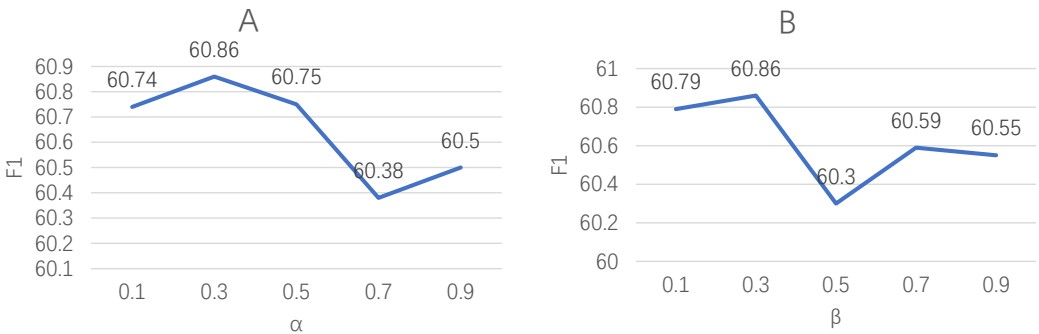

**Figure 5** These plots explore the sensitivity of our model to the hyperparameters $\alpha$ and $\beta$ in the loss function. The left plot (A) represents the variation of model performance with changes in $\alpha$, while keeping $\beta$ fixed at 0.3. Similarly, the right plot (B) depicts the impact of adjusting $\beta$ on the model performance, with $\alpha$ set as 0.3.

## CONCLUSION

ATLOP serves as the foundation for document-level relationship extraction in this work. Entity embedding representation and entity pair embedding representation are the two modules that make up the task. Using loss functions, we perform self-distillation on each module to move the backbone model's entity and entity pair representations closer to the teacher embedding representation. The base backbone model is used to approximate the upgraded teacher model during the prediction phase. Furthermore, our framework enables the training of student and teacher models simultaneously. With the DocRED document-level relational extraction dataset, our self-distilled framework successfully learns the teacher model's knowledge without growing the model size, as shown by experimental validation based on the mean of five trials.

## FUTURE WORKS

In subsequent research, we plan to investigate how more effective loss functions can enhance the self-distillation framework's performance. Furthermore, as in this article, different teacher models can have different enhancement effects by using evidence sentences and RSMAN to enhance the entity representations, respectively. However, we distill the knowledge of teacher models in this article by adding one teacher model at a time for the entity representation enhancement module and entity pair representation enhancement module. Consequently, in our next work, we aim to improve the representation of entities and entity pairs more thoroughly by simultaneously adding multiple teacher models to the entity representation enhancement module and the entity pair representation enhancement module, respectively, for distillation.

### Funding

This work was supported by the National Social Science Fund of China (No. 21BXW057). The funders had no role in study design, data collection and analysis, decision to publish, or preparation of the manuscript.

### Grant Disclosures

The following grant information was disclosed by the authors:
The National Social Science Fund of China: No. 21BXW057.

### Competing Interests

The authors declare there are no competing interests.

### Author Contributions

- Hao Wu conceived and designed the experiments, performed the experiments, analyzed the data, performed the computation work, authored or reviewed drafts of the article, and approved the final draft.
- Gang Zhou analyzed the data, prepared figures and/or tables, and approved the final draft.
- Yi Xia conceived and designed the experiments, performed the experiments, prepared figures and/or tables, and approved the final draft.
- Hongbo Liu performed the computation work, authored or reviewed drafts of the article, and approved the final draft.
- Tianzhi Zhang conceived and designed the experiments, authored or reviewed drafts of the article, and approved the final draft.

### Data Availability

The dataset and code for baselines for DocRED: A Large-Scale Document-Level Relation Extraction Dataset is available at GitHub:

https://github.com/thunlp/DocRED.

https://bitbucket.org/alexwuhkucs/gda-extraction/src/master/.

### Supplemental Information

Supplemental information for this article can be found online at http://dx.doi.org/10.7717/peerj-cs.1930#supplemental-information.

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
