# Peer review of "Self-distillation framework for document-level relation extraction in low-resource environments"

_PeerJ Computer Science, doi:10.7717/peerj-cs.1930_

## Round 0.1 · original submission · Major Revisions

Please. Pay attention to the suggestions made by the two reviewers, work on the manuscript, and submit the new revision.

**Language Note:** PeerJ staff have identified that the English language needs to be improved. When you prepare your next revision, please either (i) have a colleague who is proficient in English and familiar with the subject matter review your manuscript, or (ii) contact a professional editing service to review your manuscript. PeerJ can provide language editing services - you can contact us at copyediting@peerj.com for pricing (be sure to provide your manuscript number and title). – PeerJ Staff

Reviewer 1 ·

Basic reporting

The paper is interesting and it addresses an important topic. It contains some meaningful discussions. The references are appropriate, proposed algorithm is new and novel, and the derivations appear to be correct. The organization of the paper is smooth and well planned. Therefore, I recommend a minor revision of the paper.

Experimental design

Authors propose an algorithm for document-level relation extraction. In general, the paper has been properly written. With respect to the experiments, the algorithm has been tested in several real scenarios and its results show its usefulness and superiority over other previous approaches.

Validity of the findings

In recent years, the document-level relation extraction problem has gained attention in artificial intelligence. In order to improve the performance of the model, various tricks directly introduce additional modules into the backbone model, which usually increases the number of parameters in the entire model Although there have been several recent advances in developing algorithms for various settings of this problem, the study of generalization algorithms has been largely limited to the special applications. This paper provided a novel algorithm for this problem which fits the applications in some real-world circumstances. I recommend the paper to be accepted for publication in the PeerJ after the minor revision.

Additional comments

Minor revision is needed, since the paper has several typos. For instance, page 3 line 139, page 10, the title of table 3, ”*” should be “*”; page 4, line -2, page 6, the first line, line 165 and169, page 8, line 196, please delete two spaces before “where”. Please insert a new paragraph at the end of introduction section to state the organization of rest sections. In addition, future works should be added in the last section.

Reviewer 2 ·

Basic reporting

The study appears meaningful and well-detailed, with a sufficiently enriched experimental component. However, I would like to highlight some areas for improvement.

Does the introduction include a classification or grouping of the studies mentioned based on distinguishing points? Why have not the details been provided in the related works section?

It would be beneficial to share questions in the literature that this research seeks to answer. The method's content and its innovative aspects could be more prominently highlighted.

An outline at the end of the introduction would be a valuable addition.

In the related works section, are there more recent studies that could be included after further research? A brief explanation of how the method differs from those in related works would be helpful.

Information on coding, the used tools, and the system could be added under a separate heading.

The methods section can be confusing. Better organization and a brief summary at the beginning of this section would improve clarity.

The tables shared in the experimental results could be interpreted in greater detail.

Qualitative results could be added to the experiments for a more comprehensive analysis.

Experimental design

The experimental design is good, yet it would be beneficial if the tables provided were explained in greater detail to enhance understanding. Qualitative results could be added to the experiments for a more comprehensive analysis.

Validity of the findings

The study achieves impact and novelty.

Additional comments

-

---

## Round 0.2 · Minor Revisions

Please, pay attention to the minor suggestions made by the reviewers.

Reviewer 2 ·

Basic reporting

I have carefully reviewed the manuscript. The authors for their efforts in addressing the concerns raised in the previous reviews. The revisions made to the manuscript are successful, and they have significantly improved the overall quality of the paper.

In addition to this, I would like to suggest that the authors consider incorporating some more recent literature to further enhance the current relevance of their work.

Experimental design

The section is successfull.

Validity of the findings

The section is successfull.

·

Basic reporting

The study is significant, as it addresses a crucial issue in DocRE: efficiency. The literature references are pertinent; however, the related works section could be enhanced by providing more concise descriptions of various methods, including those later cited such as DocuNet, SIRE, and other more recents which make the SOTA.

Experimental design

The approach is innovative and well-articulated. The experiments are pertinent and properly described; however, they were conducted solely on a single dataset. Despite DocRED serving as the reference dataset in the domain, it would be valuable to evaluate this approach by testing it on at least two additional datasets (e.g., Re-DocRED, GDA, BioRED, CDR).

Validity of the findings

The study achieves novelty, and its impact could be further enhanced if the method were benchmarked on additional datasets. Therefore, I would recommend a minor revision to include these results before the paper is published.

Additional comments

Several typos should also be corrected, as well as some sentences to be rephrased:

l.34 : “More than 40.7% of associations are solely identified across many sentences, according to prior research (Yao et al., 2019” : This sentence leads me to think that these numbers concern the general domain, when Yao et al. identified this for the particular case of their dataset.
l.92: “Since the release of the transformer model in 2017 (Vaswani et al., 2017), fine-tuning large-scale pre-training models has gained widespread recognition and yielded significant improvements” : No need for this sentence.
l. 135 & l.140 : Repetition
l.145 : “Preliminary” could be renamed “Definition”. The first sentence l.146 could thus be removed.
Some typos with whitespaces l.148, l.150, l.151 and l.180.
l. 172: RSMAN : you could cite the paper here
l. 219 : Missing the Section number
l.228 : Bert = > BERT
l.233 : “Our backbone model is the ATLOP model, and we use the DocRED dataset (Yao et al., 2019) to evaluate our model’s performance" could be better worded, and leads to a repetition with l.238. Thus, you could remove the l.233.
l.270 : typo
l.275: typo, RSAMN => RSMAN
Table 4 : typo on Test
All the formulations of the type “the ATLOP model” or “the RSMAN” could just be rephrased “ATLOP” or “RSMAN”.

Reviewer 4 ·

Basic reporting

The article is interesting and address the relation extraction task using a novel method that may not outperform SOTA systems but reduces complexity which is also important in real case scenarios.
The article has already been reviewed by two experts which have done a very good work pointing out the elements that needed improvement. The authors have taken into account these remarks and have modified and added extra material.
From my point of view the article can be published after some minor modifications

Specific:

l28 "right now" -> Nowadays

l38 "connection"?

l62 "in method section" -> in the method section

l140 "we go on to describe" -> we describe

l195 - l197 -> check these sentences, they seem incomplete

l219 "In Section ?"

l235 3053 -> 3,053 . To be checked everywhere

l251 - l252 RSMAN module or RSMAN modules

l322 citation for the model is not needed, it has been cited before. To be checked everywhere

l330 "table" -> Table


General:

relationship extraction is sometimes used instead of relation extraction. Relationship and relation do not refer to the same thing

"Transformer" -> transformer
"proposes" -> propose when there is more than one author

RSMAN show be cited the first time it is mentioned on l172

Check capitalization in sections and subsections

Table captions are sentences not titles so no capitalization is needed


I suggest the authors to take a good time to revise all the article. As mentioned before there are some typos and details that need attention.

Experimental design

All comments are under Basic reporting

Validity of the findings

All comments are under Basic reporting

Additional comments

All comments are under Basic reporting

---

## Round 0.3 · accepted · Accept

The authors have addressed all the suggestions made by the reviewers.

Congrats.